# Advancing artificial intelligence applicability in endoscopy through source-agnostic camera signal extraction from endoscopic images

Ioannis Kafetzis[1]*, Philipp Sodmann[1], Robert Hüneburg[2,3], Jacob Nattermann[2,3], Nora Martens[4,5], Daniel R. Englmann[6], Wolfram G. Zoller[7], Alexander Meining[1], Alexander Hann[1]

1 Department of Internal Medicine II, Interventional and Experimental Endoscopy (InExEn), University Hospital Würzburg, Würzburg, Germany, 2 Department of Internal Medicine I, University Hospital Bonn, Bonn, Germany, 3 National Center for Hereditary Tumor Syndromes, University Hospital Bonn, Bonn, Germany, 4 Department of Medicine I, University Hospital Dresden, Dresden, Germany, 5 Else Kröner Fresenius Center for Digital Health, Technical University of Dresden, Dresden, Germany, 6 Department of Internal Medicine II, Hospital of Worms, Worms, Germany, 7 Department of Internal Medicine and Gastroenterology, Katharinenhospital, Stuttgart, Germany

* kafetzis_i@ukw.de

## Abstract

### Introduction

Successful application of artificial intelligence (AI) in endoscopy requires effective image processing. Yet, the plethora of sources for endoscopic images, such as different processor-endoscope combinations or capsule endoscopy devices, results in images that vastly differ in appearance. These differences hinder the generalizability of AI models in endoscopy.

### Methods

We developed an AI-based method for extracting the camera signal from raw endoscopic images in a source-agnostic manner. Additionally, we created a diverse dataset of standardized endoscopic images, named Endoscopic Processor Image Collection (EPIC), from 4 different endoscopy centers. Included data were recorded using 9 different processors from 4 manufacturers with 45 endoscopes. Furthermore, images recorded with 4 capsule endoscopy devices from 2 manufacturers are included. We evaluated the camera signal extraction method using 641 manually annotated images from 5 different, publicly available endoscopic image datasets, as well as on the EPIC dataset. Results were compared it with a published baseline in terms of Intersection over Union (IoU) and Hausdorff distance (HD).

### Results

In segmenting the camera signal on images from public datasets, our method achieved mean IoU of 0.97 which was significantly higher than that of the baseline

**Data availability statement:** The proposed pipeline and model weights are made available for download from https://www.kaggle.com/m/273059 (DOI: 10.34740/KAGGLE/M/273059). The EPIC dataset proposed in the paper and used for method validation can be downloaded from here https://www.kaggle.com/dsv/11103826 (DOI: 10.34740/KAGGLE/DSV/11103826). The image data for external validation are parts of five different public datasets, namely Kvasir-Insturment https://datasets.simula.no/kvasir-instrument/, PolypGen https://www.synapse.org/Synapse:syn45200214, PolypSet https://dataverse.harvard.edu/dataset.xhtml?persistentId=doi:10.7910/DVN/FCBUOR, HyperKvasir https://datasets.simula.no/hyper-kvasir/, and the El Salvador atlas gastrointestinal video endoscopy https://www.gastrointestinalatlas.com/index.html.

**Funding:** The authors AH and WGZ receive public funding from the state government of Baden-Württemberg, Germany (Funding cluster "Forum Gesundheitsstandort Baden-Württemberg") to research and develop artificial intelligence applications for polyp detection in screening colonoscopy (funding number 5409.0–001.01/15). The funders had no role in study design, data collection and analysis, decision to publish, or presentation of the manuscript.

**Competing interests:** The authors have declared that no competing interests exist.

method and mean HD of 21 pixels which was significantly lower compared to the baseline. On the standardized images of the EPIC dataset, there was no significant difference between IoU but our method achieved a significantly lower HD. Both the developed AI-based method and the generated dataset are made publicly available.

## Conclusion

This work introduces an AI-based method that effectively segments the endoscope camera signal from the raw endoscopic data in a source-agnostic way. Utilizing the proposed method as a preprocessing step allows existing AI models to use any endoscopic image, independent of its source, without compromising performance. Additionally, EPIC, a dataset of diverse endoscopic images, is generated. The proposed method, trained AI model weights, and the EPIC dataset are made publicly available.

## Introduction

In recent years, artificial intelligence (AI) has entered the medical field, finding direct involvement in patient care [1,2], showing the potential to achieve human-like performance [3]. This is particularly true for endoscopy, as the image-based nature of the examination can benefit from AI-based computer vision methods [4]. In colonoscopy, multiple AI-based systems attempt to enhance the physician's ability to detect adenoma during the examination [5–8] with several commercially computer-aided detection (CADe) systems being developed [9–13] and introduced in clinical routine [14]. Further applications of AI in endoscopy include polyp size estimation [15], characterization of colorectal polyps [16], automatic report generation for colonoscopy examinations [17], resection planning of gastrointestinal neoplasia [18], recognition of eosinophilic esophagitis [19], and management of patients suffering from Crohn's disease [20], hiatal hernias [21] and assessment of the gastroesophageal junction [22].

Raw data captured during the endoscopic examination is suboptimal to be used with AI, as the recorded interface contains the camera signal along with borders where patient and examination related information are displayed. The presence of extensive additional information in combination with the variability of the camera signal's shape and location in the raw data compromises AI performance and increases the required computational power. As a result, existing AI models cannot perform adequately "out of the box" with data from different sources. Furthermore, using uncropped images introduces model bias that diminishes model performance, as the model focuses on the borders and shape of the camera signal [23]. On the other hand, extracting only the camera signal can significantly improve model generalizability, as it vastly decreases aforementioned discrepancies in model inputs. Such a method is also beneficial for cloud computing [24], as reducing the amount of transmitted data can reduce communication delays.

Several studies evaluating AI-based systems only used preset dimensions for extracting the camera signal [5,9]. This approach might be effective on a small scale yet becomes limiting for multi-center studies which are required for proper evaluation

of AI models [25]. The lack of standardized image pre-processing and the need of manual endoscopic image cropping are acknowledged by Sierra-Jerez and colleagues in [26] as important obstacles in developing AI models in colonoscopy.

Image processing methods that can enhance performance of computer vision methods have been investigated [27]. Such methods include undistortion of the endoscopic images [28], camera calibration [29], and treatment of light artefacts and reflections [30]. Furthermore, extraction of the camera signal location has been extensively studied when the camera signal is of circular shape, which is common in endoscopic surgery [31–33]. Moving to data from gastroscopy and colonoscopy, the circularity assumption does usually not hold, as image signal is visualized in different shapes ranging from polygonal to elliptical. Mathematical methods for extracting the camera signal in these cases have been investigated [34,35]. Yet, such methods lack flexibility and thus fail to adapt to the ever-increasing diversity of raw data. AI-based cropping of endoscopic images was discussed in [23], as a method to reduce model bias. The AI presented there was not evaluated on different sources of endoscopic images and was not made publicly available.

The main aim of this work is the development and evaluation of an AI-based method that extracts the camera signal from the raw data in a source agnostic method, and to make the method and AI model weights publicly available. Such a method can support usage of AI models developed for endoscopy, by unifying input data, limiting their variability without loss of relevant information. The proposed method is evaluated using manually annotated endoscopic images from a variety of publicly available datasets. Additional evaluation is performed using a unique new dataset of standardized endoscopic images, called the Endoscopic Processor Image Collection (EPIC), which is introduced in this work. The EPIC dataset contains images recorded with a variety of sources, such as different processor-endoscope combinations and capsule endoscopy devices. Furthermore, the dataset contains endoscopic manually annotated binary masks indicating the location of the camera signal. An image processing method for performing camera signal segmentation is used as a baseline for comparison. The proposed method, AI model weights, and the EPIC dataset are made publicly available, to facilitate widespread adoption and further research in AI for endoscopy.

## Materials and methods

### Endoscopic camera signal extraction pipeline

The proposed method takes any endoscopic image as input and predicts a binary mask that segments the camera signal in the input. To reduce any prediction noise, only the largest connected component of the prediction is considered. Based on this, a minimum dimension sub-image containing the camera signal is extracted. In this work, picture-in-picture mode (PiP) is considered part of the camera signal.

The AI utilizes the UNet architecture [36] with an Efficient Net [37] as backbone. Training data for the model comprised of 1.765 manually annotated endoscopic images extracted from examination videos recorded between January 15th, 2019, and January 31st, 2022, in four different endoscopic centers. Gold standard binary semantic segmentation masks for each image were manually created to delineate the camera signal– where a value of 1 indicated pixel inclusion within the camera signal and 0 otherwise. The training data included a range of image quality levels from clear, high-quality images to low quality, blurry images. This diverse set of indicative images reflects the varied conditions encountered during endoscopy. The training images were pseudo-anonymized according to standard practice for patient data and contained no patient identifying information. An in-depth presentation of the model selection process and training details are presented in S1 File.

### Generation of the "Endoscopic Processors Image Collection" dataset

The EPIC dataset contains raw endoscopic images captured across four German hospitals with backgrounds of either white or a combination of blue and green hues. Background color selection facilitates clear identification and separation of camera signals against any borders. Example of the recording setup from one of the Hospitals is shown in Fig 1. Such images were gathered for multiple combinations of endoscopic processors and endoscopes. Given that the screen aspect

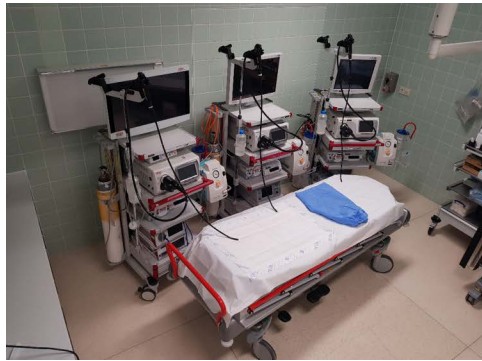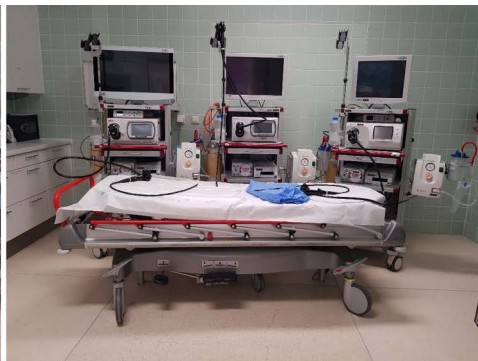

**Fig 1. Example of one of the examination room setups used in obtaining images for the EPIC dataset.** Displayed are three different Pentax (PENTAX Europe GmbH, Hamburg, Germany) processor generations with endoscopes attached. On the examination table are the two different textures used for obtaining the images.

ratio can significantly influence the final appearance of these images, various aspect ratios—16:9, 4:3, and 5:4—were also used to collect representative examples. For applications involving capsule endoscopy or cholangioscopy, where acquiring new images was not always feasible, only existing images that matched the above description and contained no patient identifying data were included in the dataset. In addition to the raw images, the EPIC dataset includes manually generated binary masks that precisely delineate the camera signal's position on each image.

No centers contributed images for both the EPIC dataset and training of the proposed AI. This guarantees that the EPIC dataset remains distinct and valid for performance evaluation purposes. For a detailed breakdown of the EPIC dataset's contents, including processor-specific data and capsule-endoscopy-related information, readers are referred to S1 Table for endoscopic processor data and S2 Table for data from capsule endoscopy.

## Method evaluation

The proposed method was evaluated on two different sets of endoscopic images. The first set consists of 641 images from five publicly available datasets [38–42] and the second set is the EPIC dataset. Gold standard was manually annotated for all images using binary masks. The mask had a value of 1 for pixels belonging to the camera signal and 0 otherwise. The method was evaluated by means of intersection over union (IoU) and Hausdorff distance (HD). IoU is defined as the percentage overlapping with the gold standard. The HD, in the context of semantic segmentation, is defined as follows. Let $m_1$ and $m_2$ be two binary masks. The one-sided HD from $m_1$ to $m_2$ is defined as $hd_s(m_1, m_2) = \max_{x \in m_1} \min_{y \in m_2} \|x - y\|_2$. This can be interpreted as the maximum distance from a point $x \in m_1$ to the closest point $y \in m_2$. The HD of $m_1$ and $m_2$ is defined as the maximum of the two one-sided HDs, that is $HD(m_1, m_2) = \max(hd_s(m_1, m_2), \ hd_s(m_2, m_1))$ [43,44]. Thus, the HD between the two masks can be seen as the greatest distance from a point in either mask to its nearest point in the other mask.

To evaluate the efficiency of our proposed method, we compared our method with the currently used standard, which is described in section 3.2 of [34], where the data pre-processing pipeline is presented. This image processing method first extracts the bright intensity from the image, compares each pixel to a threshold value and calculates the largest connected component in the image, which represents the location for the camera signal.

## Statistical analysis

Statistical analysis was performed using the SciPy [45] library for Python. The mean value and 95% confidence intervals (CI) around it are calculated for the IoU and HD. Additionally, since the measurements are continuous pairs of not

normally distributed data, the Wilcoxon test with a significance level of 0.05 is employed for comparing the proposed method and baseline performance.

## Statement of ethics

Prospective collection of endoscopic examinations during clinical routine was approved by the local ethical committee responsible for each study center (Ethik-Kommission Landesärztekammer Baden-Württemberg (F-2021–042. F-2020–158), Ethik-Kommission Landesärztekammer Hessen (2021–2531), Ethik-Kommission der Landesärztekammer Rheinland-Pfalz (2021–15677) and Ethik-Kommission University Hospital Würzburg (12/20, 20200114 04)). All procedures were in accordance with the Helsinki Declaration of 1964 and later versions. Signed informed consent from each patient where data collection was performed prospectively was obtained prior to participation.

## Results

### Composition of the EPIC dataset

The EPIC dataset comprises 267 raw endoscopic images, along with manually extracted, gold standard masks that indicate the location of the camera signal. The raw data were captured using nine different endoscopic processors: Olympus CV-180, CV-190, and CV-1500 (Olympus Europa SE & Co. KG, Hamburg, Germany), Storz Image-1S (KARL STORZ SE & Co. KG, Tuttlingen, Germany), Pentax EPK-i, EPK-i7000, and EPK-i7010 (PENTAX Europe GmbH, Hamburg, Germany), and Fujifilm VP-4450HD and VP-7000 (FUJIFILM Europe GmbH, Düsseldorf, Germany). The number of processor-endoscope combinations varied up to 23 for one included processor. Furthermore, for 14 processor-endoscope combinations, images with different aspect ratios, namely 16:9, 4:3, and 5:4, are included.

Additionally, the dataset includes images captured using four different capsule endoscopy devices from two different manufacturers: OMOM (JINSHAN Science & Technology (Group) Co., Ltd., 118 Nishang Road, Yubei, Chongqing, China) and Medtronic (MEDTRONIC TRADING NL B.V., Larixplein 4 5616 VB Eindhoven, The Netherlands). Indicative examples of endoscopic images from the EPIC dataset are presented in Fig 2.

### Evaluation of the proposed method

When tested with 641 endoscopic images from publicly available datasets, the proposed method achieved an IoU score of 0.97 (95% CI: 0.969–0.971), which was significantly higher than the mean IoU of 0.939 (95% CI: 0.932–0.946) achieved by the baseline method ($p < 0.001$). Additionally, the proposed method achieved a mean HD of 21 pixels (95% CI: 20–23), which was significantly lower than the mean HD of 51 pixels (95% CI: 45–57) achieved by the baseline ($p < 0.001$). The distributions of IoU and HD values are illustrated in Fig 3.

On the EPIC dataset, our method achieved a mean IoU of 0.962 (95% CI: 0.955–0.969), which was higher but comparable to the mean IoU of 0.954 (95% CI: 0.946–0.962) for the baseline method ($p = 0.68$). For HD, our method achieved a mean of 40 pixels (95% CI: 34–46), which was significantly lower than the mean HD of 52 pixels (95% CI: 44–60) of the baseline $p = 0.02$. The distribution of the results is shown in Fig 4, and the evaluation across different endoscopic processors in terms of IoU and HD is presented in S1 Fig.

Examples of applying the proposed method to endoscopic images from publicly available datasets are shown in Fig 5, where each row corresponds to a different test image. In the first column, the mask for the camera signal is overlaid with the image. In the second column, the results of comparing the predicted mask with the gold standard are displayed. Green color indicates true positives, red color false positives and blue color false negatives. In the third column, the camera signal is marked with a green bounding box and finally, the fourth column depicts the result of the extraction of the endoscopic image.

Finally, the time required for the proposed method to extract the camera signal was investigated. In images from public datasets, the mean execution time per image was 0.011 seconds which is close to 90 frames-per-second (fps). For images on the EPIC datasets, which are in general of higher resolution, the mean execution time per image was 0.018 seconds, or

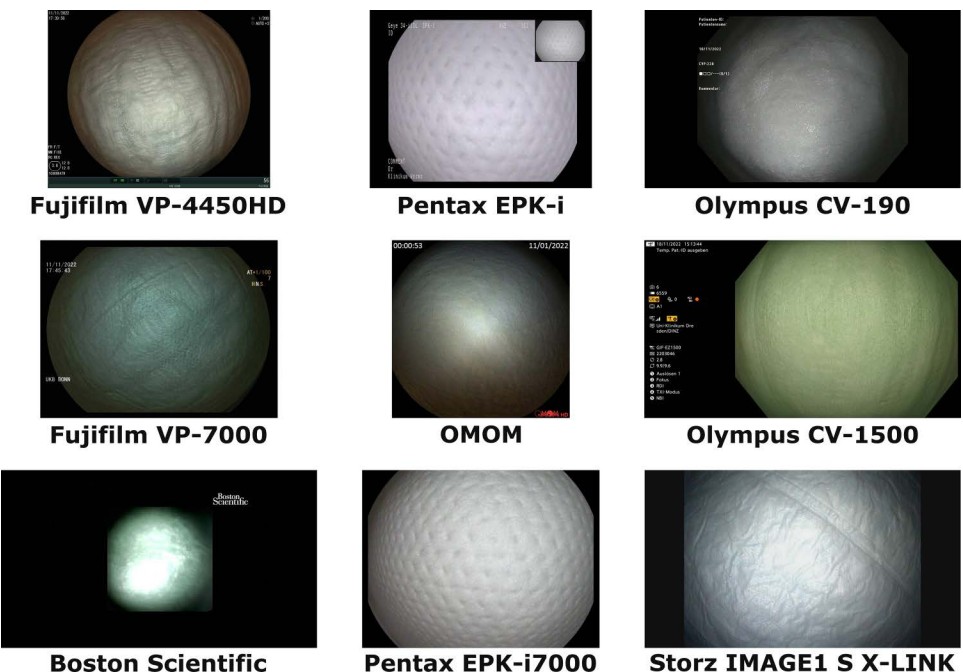

**Fujifilm VP-4450HD**  **Pentax EPK-i**  **Olympus CV-190**

**Fujifilm VP-7000**  **OMOM**  **Olympus CV-1500**

**Boston Scientific**  **Pentax EPK-i7000**  **Storz IMAGE1 S X-LINK**

**Fig 2. Image examples of images from the EPIC dataset with their capturing devices.** The images are displayed in their original resolution.

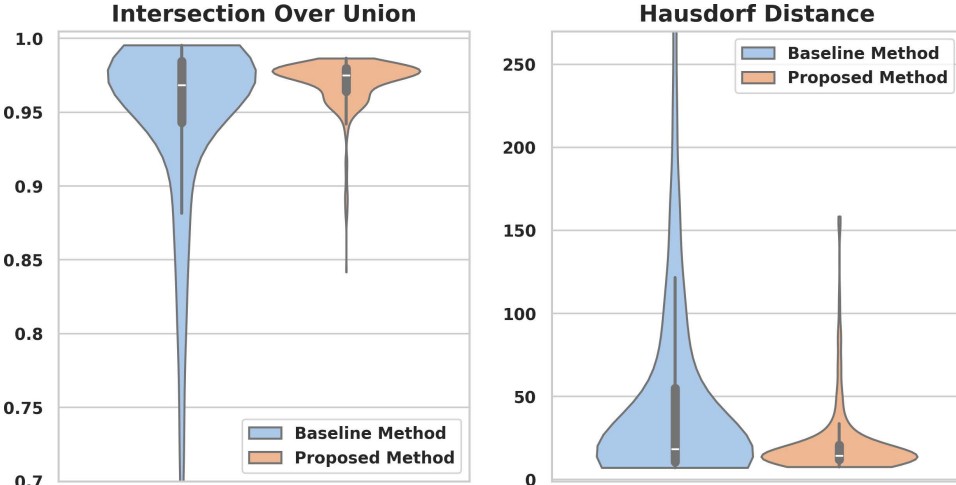

**Fig 3. Performance comparison of the proposed and baseline methods when extracting the endoscopic camera signal for images from public datasets.** The distributions of intersection over union (left) and Hausdorff distance (right) values on the test dataset are compared.

52 fps. The results were obtained using an NVIDIA GeForce RTX 3080 Ti (NVIDIA Corporation, 2788 San Tomas Expressway, Santa Clara, CA 95051, USA). Both cases indicate that the model can achieve real-time performance.

## Discussion

Although AI methods have been presented as highly beneficial for physicians, especially in the field of endoscopy, it remains challenging to integrate these models into clinical settings due to significant variability among endoscopic images

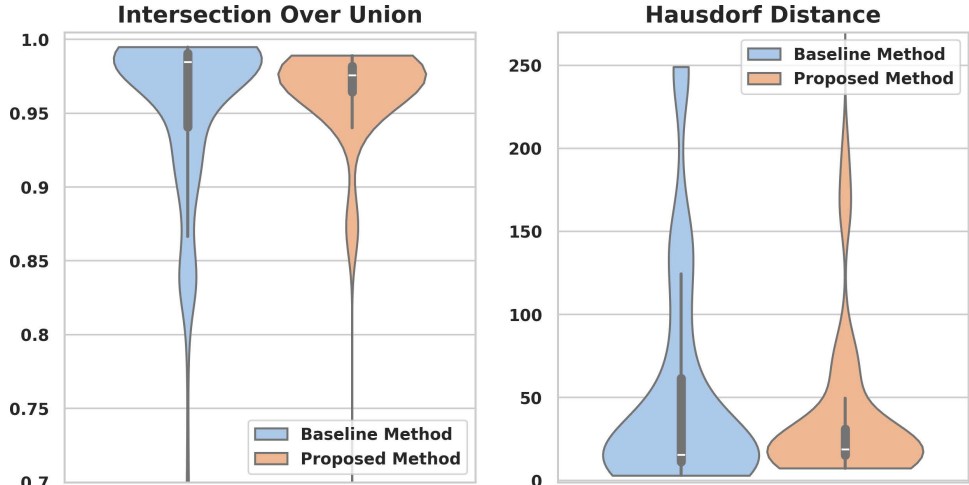

**Fig 4. Performance comparison of the proposed and baseline methods when extracting the endoscopic camera signal from images of the EPIC dataset.** The distributions of intersection over union (left) and Hausdorff distance (right) values on the test dataset are compared.

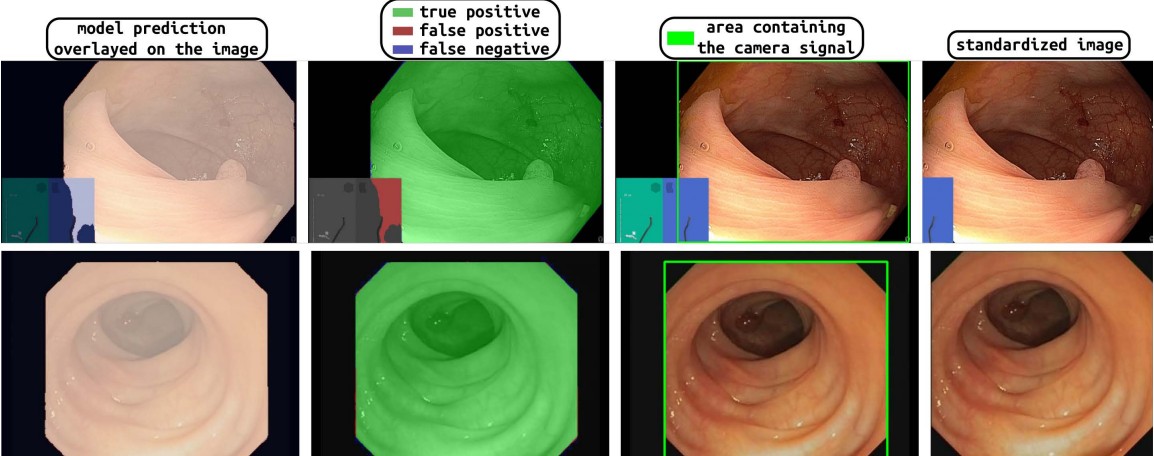

**Fig 5. Examples of application of the proposed method to images from the test dataset.** Each row corresponds to a different example. The first column overlays the predicted mask on the original image. The second column evaluates the predicted mask, with green color indicating true positives, red color false positives and blue color false negatives. The third column displays the original image, and the green box indicates the minimum sub-image containing the camera signal, as obtained with the proposed method. The last column displays the extracted endoscopic image.

used as input. This variability is primarily attributed to differences in the hardware, such as endoscopic processors and endoscopes, and software such as the device used to record the examination.

The main contribution of this work is the development of an AI-based method that, given raw endoscopic image data as input, efficiently extracts the location of the endoscopic camera image in a manner that is agnostic to the source, that is independent of the specific endoscopic hardware or software used to capture the images. The generated model and trained weights are made publicly available under https://www.kaggle.com/m/273059 (https://doi.org/10.34740/KAG-GLE/M/273059), to support broader adoption from the research community.

Furthermore, we created a dataset of raw endoscopic images called the EPIC dataset. Our goal with this data-set is threefold: to highlight the substantial diversity present in endoscopic images, to determine the causes for their

variability, and to catalog these diverse images in a standardized format. The EPIC dataset includes 267 raw endoscopic data captured using nine different processors, multiple different endoscopes, and four different capsule devices used for capsule endoscopy. The variety of combinations between hardware and recording options used makes the proposed dataset ideal for illustrating the wide range of shapes and forms in which the camera signals in displayed in raw endoscopic images. Moreover, the EPIC dataset contains manually annotated binary masks indicating the location of the camera signal for each image. The diversity of the dataset can be further highlighted considering the time-span of endoscopic images covered, with the oldest endoscopic processor included was introduced in 2006 whereas the newest one was launched in 2020.The EPIC dataset is made available under https://doi.org/10.34740/kaggle/dsv/11103826

The proposed method was evaluated using images from publicly available endoscopic datasets and the EPIC dataset. In segmenting the camera signal from public dataset images, our AI achieved a mean IoU 0.97 and mean HD of just 21 pixels, which are significantly better compared to the mean IoU of 0.939 ($p < 0.01$) and mean HD of 51 pixels ($p < 0.01$) that the baseline achieved. On the EPIC dataset, the proposed method achieved a mean IoU of 0.962 which was comparable to the 0.954 of the baseline ($p = 0.68$) and HD of 40 pixels which was significantly lower than the 52 pixels of the baseline ($p = 0.02$).The images of the EPIC dataset highly benefit the baseline method, as they are captured so that the camera signal has a color easily distinguishable from the background. Therefore, the baseline method was expected to achieve top results when evaluated on the EPIC dataset. Furthermore, the AI model has never encountered similar looking images during its training process.

Robust and reliable automatic extraction of the endoscopic camera signal from the raw data is necessary in the development of AI for colonoscopy [26]. In this direction, Yao H. and colleagues performed the cropping of the endoscopic image by binarizing it and detecting the largest 4-connected component [34], which served as a baseline for comparison of the proposed method. In [35] endoscopic images are cropped by first converting them in grayscale and then extracting a circular region with its center matching the center of the image. This method results, by design, in the loss of information from the endoscopic image. Finally, the work closest to our results is that of [23], where a U-Net model is used to determine the location of the camera signal in the endoscopic image. Yet, the work did not evaluate the model performance, especially how it performs on images from different sources and did not make their model available to the public. In terms of creating images with masks indicating the camera signal, Sanchez-Peralta et. al. proposed a dataset of raw endoscopic images, where masks for the camera signal are also provided [46]. Yet, all the images come from one processor, and extraction of the mask for the endoscopic image is manually performed. Finally, several works effectively extract the endoscopic camera signal for surgical videos, where the part of the image containing the signal is assumed to be circular [31–33]. Yet, the assumption of a circular camera signal is not present in the general endoscopic images, where the shape ranges from ellipse to polygonal.

We believe that our method can significantly enhance the generalizability of AI models trained for endoscopy. By integrating the proposed method into existing and future AI models, we achieve a streamlined process. First, raw endoscopic images are processed with our method to extract only the camera signal. This contains all relevant information necessary for the AI model while minimizing irrelevant data, thereby standardizing the input images. The standardized image is then used as input to the AI. This way, AI input includes all relevant and only a minimum amount of irrelevant information, independent from their source. This further contributes to standardizing endoscopic images, a process that has been already shown to significantly impact AIs, for example in esophageal cancer detection [47] and improved mucosal visualization in capsule endoscopy [48]. Thus, introduction of our method can improve generalizability of AI for endoscopy to data from any sources without any additional overhead. Furthermore, we believe that the proposed method can easily find successful application in pipelines for pre-processing endoscopic image data and AI model training. The fact that the image area to be cropped is selected as the minimum area rectangle containing all the mask pixels, together with the high performance of the AI suggests that no loss of relevant information occurs from using the proposed method.

Inclusion of data from different sources has already proved beneficial in training AI models that find applications in clinical practice. As an example, diversity in data training allowed successful application of an AI in clinical routine in multiple different endoscopy centers in [49], achieving high performance despite usage of different hardware. Furthermore, studies have incorporated images from publicly available datasets as external validation, where standardization of images from different sources plays a central role [21]. Finally, our proposed method can address the challenge of varying endoscopic equipment across different centers, facilitating the execution of multicenter studies for AI in endoscopy.

Endoscopy is a rapidly developing field, especially in terms of hardware. Furthermore, there are certainly sources of data that are rare, and thus obtaining relevant images can be harder. Considering this, the ability of the proposed method to remain performant when used with data from newer devices is significant. We are confident that the model can maintain high performance, as its evaluation was performed on two datasets that had no overlap with the training dataset. Furthermore, we make the model publicly available and welcome data submission from researchers to further develop and improve model performance. Any newer versions of the model weights will also be made available, acknowledging all data contributions as well.

There are also some limiting factors for this work. Endoscopic data usually contains elements such as motion blur and artefacts, that could be disrupting to the AIs performance. To mitigate this problem, such data were also annotated and included in the training dataset. Furthermore, in the case of video data, the cropping dimensions can be obtained via averaging of sequential frames, enabling removal of any outliers. Another limitation is that the EPIC dataset does not cover the whole spectrum of existing combinations. To address this issue, we plan to keep updating the EPIC dataset, as well as welcome and acknowledge image contributions that extend it.

## Conclusions

In this work, we propose an AI-based method that effectively extracts camera signals from raw endoscopic data, independent of the endoscopy hardware and software used to record them. This can enhance the standardization of images used as input to AI models, thereby increasing their transferability and generalizability across diverse clinical settings. This is crucial for maintaining consistent, high-quality data with reduced variability issues that can arise when training AI models on diverse datasets. Moreover, the proposed method's source agnosticism supports data sustainability by unifying diverse datasets into standardized format, simplifying their inclusion in AI training and evaluation pipelines.

Additionally, we generate a dataset, called EPIC, of standardized endoscopic images, attempting to highlight and collectively report the diversity that can be introduced by different endoscopic equipment.

By making both the proposed method and the EPIC dataset publicly available, we aspire to generate a collaborative environment where researchers can build upon these foundational resources to further advance their work in AI-driven endoscopic applications.

## Supporting information

**S1 Table. Image data from endoscopic processors included in the EPIC dataset.** The EPIC dataset contains 267 images stored using 9 different endoscopic processors, several endoscopes and different recording aspect ratio settings. ERCP: Endoscopic Retrograde Cholangiopancreatography, EPIC: Endoscopic Processor Image Collection.
(DOCX)

**S2 Table. Image data from capsule endoscopy included in the EPIC dataset.** Description of the different small bowel and colon video capsules included in the EPIC dataset. EPIC: Endoscopic Processor Image Collection.
(DOCX)

**S1 Fig. Performance of the proposed and baseline methods on the different endoscopic processors.** The subgroup analysis for each processor in terms of intersection over union and Hausdorff distance on the different processors

included in the EPIC dataset is displayed. The mean value is indicated with a circle and lines depict 95% confidence intervals.
(TIFF)

**S1 File. Camera signal extraction pipeline and training of the camera image segmentation model.** Description of the pipeline and training for the proposed method.
(DOCX)

## Author contributions

**Conceptualization:** Ioannis Kafetzis, Alexander Hann.

**Data curation:** Ioannis Kafetzis, Robert Hüneburg, Jacob Nattermann, Nora Martens, Daniel R. Englmann, Wolfram G. Zoller, Alexander Meining, Alexander Hann.

**Formal analysis:** Ioannis Kafetzis.

**Investigation:** Ioannis Kafetzis.

**Methodology:** Ioannis Kafetzis, Philipp Sodmann, Alexander Hann.

**Resources:** Robert Hüneburg, Jacob Nattermann, Nora Martens, Daniel R. Englmann, Wolfram G. Zoller.

**Software:** Ioannis Kafetzis, Philipp Sodmann.

**Supervision:** Alexander Meining, Alexander Hann.

**Validation:** Ioannis Kafetzis.

**Visualization:** Ioannis Kafetzis, Philipp Sodmann.

**Writing – original draft:** Ioannis Kafetzis.

**Writing – review & editing:** Ioannis Kafetzis, Philipp Sodmann, Robert Hüneburg, Jacob Nattermann, Nora Martens, Daniel R. Englmann, Wolfram G. Zoller, Alexander Meining, Alexander Hann.

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
