## [Decision Letter · Decision Letter 0]

Dear Dr. Kafetzis,

Thank you for submitting your manuscript to PLOS ONE. After careful consideration, we feel that it has merit but does not fully meet PLOS ONE’s publication criteria as it currently stands. Therefore, we invite you to submit a revised version of the manuscript that addresses the points raised during the review process.

**ACADEMIC EDITOR:**

Minor enhancements to the discussion of broader impacts and potential limitations could provide a more rounded perspective.

We look forward to receiving your revised manuscript.

Kind regards,

Kazunori Nagasaka

Academic Editor

PLOS ONE

Journal Requirements:

3. Thank you for stating the following financial disclosure: [The authors Alexander Hann and Wolfram G. Zoller receive public funding from the state government of Baden-Württemberg, Germany (Funding cluster “Forum Gesundheitsstandort Baden-Württemberg”) to research and develop artificial intelligence applications for polyp detection in screening colonoscopy (funding number 5409.0–001.01/15).]. 

5. We note that you have indicated that there are restrictions to data sharing for this study. For studies involving human research participant data or other sensitive data, we encourage authors to share de-identified or anonymized data. However, when data cannot be publicly shared for ethical reasons, we allow authors to make their data sets available upon request. For information on unacceptable data access restrictions, please see http://journals.plos.org/plosone/s/data-availability#loc-unacceptable-data-access-restrictions. 

Additional Editor Comments:

Dear Authors,

Thank you for submitting your manuscript to Plos One.

Our expert reviewers have recommended to revise the manuscript.

Please revise the manuscript accordingly.

I think the manuscript addresses a significant issue in the application of AI in medical imaging.

Minor enhancements to the discussion of broader impacts and potential limitations could provide a more rounded perspective.

Best regards,

Kazunori Nagasaka

Reviewers' comments:

Reviewer's Responses to Questions

**Comments to the Author**

1. Is the manuscript technically sound, and do the data support the conclusions?

Reviewer #1: Partly

Reviewer #2: Yes

2. Has the statistical analysis been performed appropriately and rigorously?

Reviewer #1: No

Reviewer #2: Yes

3. Have the authors made all data underlying the findings in their manuscript fully available?

Reviewer #1: No

Reviewer #2: Yes

4. Is the manuscript presented in an intelligible fashion and written in standard English?

Reviewer #1: No

Reviewer #2: Yes

Reviewer #1: This manuscript proposed a camera signal extraction pipeline in endoscopy by using deep learning UNet model. Definitely the topic is interesting and important. My comments are:

1. The manuscript was poorly organised and written. English is poor, hard to follow. The quality of the figures are very low, even without labels at all.

2. There exist lots of studies in AI for endoscopic image/video analysis as summarise in https://doi.org/10.1038/s41746-022-00733-3. Why do we need another one like the one in this manuscript, given that “the main contribution of this work is the development of an AI-based method that, given raw endoscopic image data as input, manages to extract, in a source agnostic way, the location of the endoscopic camera image from the raw data.”? – I do not see any new developments.

3. Throughout the manuscript, authors talked about baseline models, but I am struggling to find what it is. It seemed to mention the paper by Yao (23) but that paper is about motion-based camera localization system in colonoscopy videos rather than simply segmenting images.

4. The results simply showed IoU and HD for the proposed method and “baseline method” – The analysis of effects of multi-centres and multi-camera devices is lacking.

5. In conclusion, the authors claimed that “This method can be used with any existing AI model to improve its ability to generalize to data from different sources, thus increasing transferability and sustainability.” – it is hardly convincing.

Reviewer #2: The study assesses the AI-based strategy utilizing images from nine distinct processors and four capsule endoscopy equipment; yet, endoscopic technology is perpetually advancing. Could the authors elaborate on the efficacy of their strategy when applied to newly designed or less frequently utilized endoscopic devices? What strategies might be employed to mitigate this constraint in further research?

Although the method attains elevated IoU and diminished Hausdorff distance on the EPIC dataset and public datasets, the study fails to address its efficacy in real-time clinical environments characterized by dynamic elements such as motion blur, lighting fluctuations, and aberrations. Could the authors address the possible limits of implementing this strategy in a clinical workflow and suggest measures to enhance its robustness?

The research contrasts the suggested method with only one published baseline. Could the authors elucidate the rationale for the exclusion of other existing AI-based segmentation algorithms from the comparative analysis? In what manner does this omission influence the study's capacity to thoroughly validate its superiority?

The article asserts that the proposed strategy enables AI models to generalize across various endoscope sources. Was this assertion corroborated by training and evaluating an independent AI model on preprocessed photos from other sources? Otherwise, may the authors elaborate on the importance of performing such an evaluation?

The standardization of images from many sources may adversely affect nuanced image characteristics essential for specific AI-driven diagnostic applications. Can the authors address whether their strategy modifies clinically significant image details? How can this constraint be mitigated?

The authors must articulate a more compelling rationale for this study by detailing the imperative of creating a source-agnostic AI-driven segmentation technique for endoscopic pictures. The study effectively handles the issue of image variability from various endoscopic sources; nonetheless, it would benefit from a discussion on the role of standardization and preprocessing in enhancing AI-based medical image interpretation. Recent studies, including Chen et al. (2025), which assessed band selection strategies for improved visualization in esophageal cancer detection, and Wang et al. (2024), which illustrated the significance of hyperspectral imaging in augmenting mucosal visualization during capsule endoscopy, underscore the necessity of refining image processing techniques to elevate diagnostic precision. Referencing these works would enhance the rationale for the proposed method and situate its contribution within the wider progress in AI-enhanced endoscopic imaging:

1. Chen, Yen-Chun, et al. "Evaluation of Band Selection for Spectrum-Aided Visual Enhancer (SAVE) for Esophageal Cancer Detection." Journal of Cancer 16, no. 2 (2025): 470.

2. Wang, Yen-Po, et al. "Spectrum aided vision enhancer enhances mucosal visualization by hyperspectral imaging in capsule endoscopy." Scientific Reports 14, no. 1 (2024): 22243.

**Do you want your identity to be public for this peer review?** For information about this choice, including consent withdrawal, please see our Privacy Policy

Reviewer #1: No

Reviewer #2: **Yes: ** Hsiang-Chen Wang

---

## [Author Response · Author response to Decision Letter 1]

20 Mar 2025

Dear Professor Kazunori Nagasaka,

We would like to express our sincere gratitude for your thoughtful consideration and

insightful comments on our manuscript. We have carefully addressed your suggestions and

made the necessary revisions to enhance the clarity and impact of our work.

As recommended, we have expanded the discussion section to include a detailed

explanation of how our proposed method can be integrated into both the development and

application of AI in endoscopy. We have also highlighted how this integration can improve

data preprocessing pipelines and AI model training for endoscopic image analysis. It now

reads;

“We believe that our method can significantly enhance the generalizability of AI models

trained for endoscopy. By integrating the proposed method into existing and future AI

models, we achieve a streamlined process where raw endoscopic images are first

processed through our method to extract only the camera signal, which contains all relevant

information necessary for the AI model while minimizing irrelevant data, thereby

standardizing the input images. This way, the images for the AI include all relevant

information, a minimal amount of irrelevant information, and are as standardized as possible,

independent of their source. This further contributes to standardizing endoscopic images, a

process that has already been shown to significantly impact AI performance, for example, in

esophageal cancer detection (43) and improved mucosal visualization in capsule endoscopy

(44). Thus, the introduction of our method can greatly benefit the model’s ability to

generalize data from any source without additional overhead. Furthermore, we believe that

the proposed method can easily find successful application in pipelines for pre-processing

endoscopic image data and AI model training. The fact that the image area to be cropped is

selected as the minimum area rectangle containing all the mask pixels, together with the

high performance of the AI, suggests that no loss of relevant information occurs from using

the proposed method.”

We believe this expanded discussion better illustrates the potential impact of our method on

clinical practice and future AI research in endoscopy. In addition, based on reviewer

feedback, we have incorporated an analysis of execution time in the results section and

extended the discussion to address potential limitations. This includes consideration of

newer endoscopic hardware and the impact of endoscopic image content on our method’s

performance. By incorporating these points, we aim to provide a more balanced perspective

on the strengths and potential challenges of our approach.

Lastly, we have ensured that our manuscript aligns with the journal's required formatting

style. This also included and updated version of the financial disclosure which now

additionally states that “The funders had no role in study design, data collection and

analysis, decision to publish, or preparation of the manuscript”.We believe these revisions align with your expectations, and we greatly appreciate your

support. Please find below our detailed, point-by-point response to each reviewer comment.

I remain at your disposal for any further inquiries.

With kind regards, on behalf of all the authors,

Dr. Ioannis Kafetzis

Response to Reviewer 1 comments:

This manuscript proposed a camera signal extraction pipeline in endoscopy by using deep

learning UNet model. Definitely the topic is interesting and important.

Response:

We thank you for your acknowledgment that the topic of the manuscript on a camera signal

extraction pipeline in endoscopy using deep learning UNet models is important and

interesting.

Comment 1:

The manuscript was poorly organised and written. English is poor, hard to follow. The quality

of the figures are very low, even without labels at all.

Response:

We thank the reviewer for their feedback. We have thoroughly revised the paper to enhance

expression and clarity. This included seeking help from a native English speaker to refine the

language. Based on their feedback, we believe that the updated version is clearer.

Additionally, we restructured the order in which points are presented in the methods section

to facilitate a more intuitive reading experience. To ensure high-quality visuals, we generated

all figures again at a resolution of 300 dpi to enhance clarity and detail. In Figure 5, we have

added labels above each column of images to make the contents of each image clearer.

With these revisions, we aim to improve the quality and readability of the manuscript.

Comment 2:

There exist lots of studies in AI for endoscopic image/video analysis as summarise in

https://doi.org/10.1038/s41746-022-00733-3. Why do we need another one like the one in

this manuscript, given that “the main contribution of this work is the development of an AI-

based method that, given raw endoscopic image data as input, manages to extract, in a

source agnostic way, the location of the endoscopic camera image from the raw data.”? – I

do not see any new developments.

Response:

We thank the reviewer for their insightful comments and for highlighting that our work did not

clearly articulate its main contributions. We have included the provided review andcharacteristic references in our introduction to better contextualize our research. Additionally,

we added more details about related methods attempting to extract camera signals from

endoscopic images under the assumption of a circular shape, pointing out the need to

extend these methos as camera signal in gastroscopies and colonoscopies do not typically

appear in this form.

We believe this context makes it clear why we developed a source-agnostic method for

extracting the endoscopic camera. Furthermore, we mention previous works attempting a

similar approach to ours, but without extensive evaluation or making the model publicly

available.

Comment 3:

Throughout the manuscript, authors talked about baseline models, but I am struggling to find

what it is. It seemed to mention the paper by Yao (23) but that paper is about motion-based

camera localization system in colonoscopy videos rather than simply segmenting images.

Response:

We appreciate the careful review and have revised the manuscript to address the feedback.

Indeeed the paper of Yao (2023) is focused on motion-based camera localization in

colonoscopy videos. Yet, in their section for data preparation they explicitly state the steps

followed to extract the camera signal. We have made updated the description in our

manuscript to pinpoint where the method is described in the original work. The passage now

reads;

“To evaluate the efficiency of our proposed method, we compared our method with the

currently used standard, which is described in (30), in the section 3.2 analyzing the data pre-

processing pipeline.”

Comment 4:

The results simply showed IoU and HD for the proposed method and “baseline method” –

The analysis of effects of multi-centres and multi-camera devices is lacking.

Response:

We appreciate your comments and have revised address the issues you pointed out.

The reviewer is right to point out that the manuscript was missing a comparison of the

impact that different hardware had. To address this, we added a subgroup analysis per

endoscopic processor for IoU (Intersection over Union) and HD in the supplementary

section, providing a more detailed comparison of the proposed method and the baseline

method.

The reviewer also mentioned multi-camera devices. Unfortunately, the available

endoscopes, we were only able to use standard monocular devices. Therefore, we did not

have any information on multi-camera devices available in this study.Comment 5:

In conclusion, the authors claimed that “This method can be used with any existing AI model

to improve its ability to generalize to data from different sources, thus increasing

transferability and sustainability.” – it is hardly convincing.

Response:

The reviewer is correct to point out that the conclusion is not convincing. To address this, we

re-wrote the entire conclusion section and now reference specific applications of our method

which demonstrate improved transferability and data sustainability. With these references,

we believe that our claims are now better supported.

Here is the revised conclusion:

“In this work, we propose an AI-based method that effectively extracts camera signals from

raw endoscopic recording data recording with different hardware and software options. This

can enhance the standardization of images used as input to AI models, thereby increasing

their transferability and generalizability across diverse clinical settings. This is crucial for

maintaining consistent, high-quality data with reduced variability issues that can arise when

training AI models on diverse datasets. Moreover, the proposed method's source

agnosticism supports data sustainability by unifying diverse datasets into standardized

format, simplifying their inclusion in AI training and evaluation pipelines.

Additionally, we generate a dataset, called EPIC, of standardized endoscopic images,

attempting to highlight and collectively report the diversity that can be introduced by different

endoscopic equipment.

By making both the proposed method and the EPIC dataset publicly available, we aspire to

generate a collaborative environment where researchers can build upon these foundational

resources to further advance their work in AI-driven endoscopic applications.”

Response to Reviewer 2 comments:

Comment 1:

The study assesses the AI-based strategy utilizing images from nine distinct processors and

four capsule endoscopy equipment; yet endoscopic technology is perpetually advancing.

Could the authors elaborate on the efficacy of their strategy when applied to newly designed

or less frequently utilized endoscopic devices? What strategies might be employed to

mitigate this constraint in further research?

Response:

The reviewer is right to point out that handling of data from newer devices has been noted in

the discussion section. We acknowledge that endoscopic technology is rapidly advancing,

and our AI-based strategy was tested with images from nine distinct processors and four

capsule endoscopy equipment. However, we recognize that there are newer and less

frequently utilized endoscopic devices that may pose challenges.To address this constraint, we propose several strategies for future research:

Data Collection: We acknowledge the difficulty in obtaining relevant data from newer and

higher-quality devices. To mitigate this, we encourage researchers to submit their datasets,

which will help us update our model accordingly.

Model Generalization: Our AI-based strategy was validated on distinct and external

datasets. Given its architecture, it is designed to maintain high performance even when

applied to new data sources.

Public Release of Updated Models: We make the proposed method publicly available and

welcome contributions from researchers who can provide additional datasets. Any updated

versions of our model will be made available to ensure continued improvement and

relevance.

“Endoscopy is a rapidly developing field, especially in terms of hardware. Furthermore, there

are certainly sources of data that are rare, and thus obtaining relevant images can be

harder. Considering this, the ability of the proposed method to perform on data from newer

devices is significant. We are confident that the model can maintain high performance, as its

evaluation was performed on two datasets that had no overlap with the training dataset.

Furthermore, we make the model publicly available and welcome data submission from

researcher to further develop and improve model performance. Any updated to the model

will be also made available, acknowledging data contributions as well.”

Comment 2:

Although the method attains elevated IoU and diminished Hausdorff distance on the EPIC

dataset and public datasets, the study fails to address its efficacy in real-time clinical

environments characterized by dynamic elements such as motion blur, lighting fluctuations,

and aberrations. Could the authors address the possible limits of implementing this strategy

in a clinical workflow and suggest measures to enhance its robustness?

Response:

We appreciate the reviewer’s comments and understand the importance of evaluating the

method's efficacy in real-time clinical environments. The following sections provide additional

details on the implementation limits and measures to enhance robustness. The reviewer

noted that the implementation in real-time was not explicitly mentioned, which is indeed a

critical aspect for practical application. To address this concern, we have evaluated the

mean time required for the model to process test data and translate it into frames per second

(fps).

This analysis is included in the results section of our paper. In the worst-case scenario, the

AI achieved a performance of 52 fps, which is more than satisfactory for incorporating into

clinical workflows. The passage in the results section reads:

“Finally, the time required for the proposed method to extract the camera signal was

investigated. In images from public datasets, the mean execution time per image was 0.011

seconds which is close to 90fps. For images on the EPIC datasets, which are in general ofhigher resolution, the mean execution time per images was 0.018 seconds, or 52 fps. The

results were obtained by using the method with an NVIDIA GeForce RTX 3080 Ti (NVIDIA

Corporation, 2788 San Tomas Expressway, Santa Clara, CA 95051, USA). Both cases

indicate that the model can achieve real-time performance.”

The reviewer is also correct to highlight that low image quality data could impact model

performance. We took this into account while developing the model, training with both high-

and low-quality data. We now explicitly point this out in the corresponding methods section,

which reads:

“The AI follows the UNet architecture (34) with an Efficient Net (35) as backbone. Training

data for the model consisted of 1.765 manually annotated endoscopic images extracted from

examination videos recorded between January 15th, 2019, and January 31st, 2022, in four

different endoscopic centers. The training data included a range of image quality levels from

clear, high-quality images to lower quality, blurry images. This diverse set of indicative

images reflects the varied conditions encountered during endoscopy.”

Finally, to address real time use, we added in the discussion section the following passage:

“Endoscopic data usually contain elements such as motion blur and artefacts, that could be

disrupting to the AIs performance. To mitigate this problem, such data were also annotated

and included in the training dataset. Furthermore, in case of video data, the cropping

dimensions can be obtained via averaging of sequential frames, enabling removal of any

outliers.”

Comment 3:

The research contrasts the suggested method with only one published baseline. Could the

authors elucidate the rationale for the exclusion of other existing AI-based segmentation

algorithms from the comparative analysis? In what manner does this omission influence the

study's capacity to thoroughly validate its superiority?

Response:

In the introduction, we now clearly state that our research contrasts the suggested method

with only one publicly available baseline due to the specific focus on gastroenterology

applications where circular endoscopic images are extracted, which is not a common

scenario for other AI-based segmentation algorithms. This explains why certain existing

methods were excluded from the comparison.

Additionally, in the introduction and conclusion section, we have emphasized that a recent

paper used a UNET architecture for camera signal segmentation but did not provide any

model or checkpoint, making it impossible to compare their results directly. Therefore, this

method was also excluded from our comparative analysis.

Comment 4:

The article asserts that the proposed strategy enables AI models to generalize across

various endoscope sources. Was this assertion corroborated by training and eva

---

## [Decision Letter · Decision Letter 1]

Dear Dr. Kafetzis,

Thank you for submitting your manuscript to PLOS ONE. After careful consideration, we feel that it has merit but does not fully meet PLOS ONE’s publication criteria as it currently stands. Therefore, we invite you to submit a revised version of the manuscript that addresses the points raised during the review process.

We look forward to receiving your revised manuscript.

Kind regards,

Kazunori Nagasaka

Academic Editor

PLOS ONE

Additional Editor Comments (if provided):

Dear Authors,

Thank you for your revised manuscript submission. After careful review, the manuscript still requires substantial improvements. Specific concerns include:

Abstract Format: The abstract, particularly the last line, is improperly formatted. Please carefully proofread and format according to the journal's guidelines.

Figures Missing: Figures 1 and 2 referenced in the text cannot be located. Ensure all figures referenced are clearly included and properly labeled.

Reference Accuracy: Reference 21 lacks essential publication information. Please verify and complete all references to comply with journal standards.

Technical Definition: The manuscript's definition of "Hausdorff distance (HD) as the maximum distance between predictions and the gold standard" is overly simplistic. Provide a more precise and technically accurate definition appropriate for a scholarly audience.

Gold Standard Generation: It remains unclear how the labels or gold standards for training were generated. Clarify your methodology for creating these standards to ensure reproducibility and transparency.

Technical Contribution: The manuscript currently employs a UNet model without notable innovations or modifications. To enhance technical value, clearly state and elaborate on novel aspects or significant modifications of your approach compared to existing methodologies.

Due to these critical concerns, your manuscript requires major revisions. Please address each point thoroughly and resubmit a significantly improved version.

Sincerely,

Kazunori Nagasaka

Reviewers' comments:

Reviewer's Responses to Questions

**Comments to the Author**

Reviewer #1: All comments have been addressed

Reviewer #2: All comments have been addressed

2. Is the manuscript technically sound, and do the data support the conclusions?

Reviewer #1: No

Reviewer #2: Yes

3. Has the statistical analysis been performed appropriately and rigorously?

Reviewer #1: Yes

Reviewer #2: Yes

4. Have the authors made all data underlying the findings in their manuscript fully available?

Reviewer #1: Yes

Reviewer #2: Yes

5. Is the manuscript presented in an intelligible fashion and written in standard English?

Reviewer #1: No

Reviewer #2: Yes

Reviewer #1: The revised manuscript is far from satisfactory. The quality of the production is still quite low, say, the last line of the abstract is in improper format, there is no where to find Figure 1 and Figure 2, reference 21 has no publication information etc. to name a few. Moreover, you cannot simply say "Hausdorff distance (HD) is the maximum distance between predictions and the gold standard." The design of the method in the manuscript is not clear how you generate the labels/gold standard for training. More importantly, the manuscript simply applied the UNet rather than developed some new techniques, which show very little technical contributions.

Reviewer #2: The authors all reply my comments. I haven't no more comments. This article can be accepted by PLOS One.

**Do you want your identity to be public for this peer review?** For information about this choice, including consent withdrawal, please see our Privacy Policy

Reviewer #1: No

Reviewer #2: **Yes: ** Hsiang-Chen Wang

---

## [Author Response · Author response to Decision Letter 2]

2 May 2025

Dear Prof. Kazunori Nagasaka,

We would like to sincerely thank you for considering our manuscript for publication. We also apologize for the oversight regarding the missing figures in the revised manuscript. Please rest assured that we have carefully addressed all your comments, as well as those of the reviewers, in this revised version.

Below, we outline the actions taken in response to your specific concerns:

Comment 1: Abstract Format: The abstract, particularly the last line, is improperly formatted. Please carefully proofread and format according to the journal's guidelines.

Response: We apologize for the misalignment caused by applying text justification. This has now been corrected, and the abstract is properly aligned and formatted according to the journal's guidelines.

Comment 2: Figures Missing: Figures 1 and 2 referenced in the text cannot be located. Ensure all figures referenced are clearly included and properly labeled.

Response: We sincerely apologize for this oversight. In the revised submission, all figures are now properly included, clearly labeled, and meet the journal’s high-quality standards, with a resolution of 300 dpi.

Comment 3: Reference Accuracy: Reference 21 lacks essential publication information. Please verify and complete all references to comply with journal standards.

Response: We have updated Reference 21 by adding the missing publication details, including the proceedings in which the work was published, to ensure compliance with the journal’s referencing requirements.

Comment 4: Technical Definition: The manuscript's definition of "Hausdorff distance (HD) as the maximum distance between predictions and the gold standard" is overly simplistic. Provide a more precise and technically accurate definition appropriate for a scholarly audience.

Response: We recognize that the original definition was too simplistic. In the revised manuscript, we have included the formal mathematical definition of the Hausdorff distance and expanded the explanation to provide a more accurate and comprehensive description suitable for a scholarly audience.

Comment 5: Gold Standard Generation: It remains unclear how the labels or gold standards for training were generated. Clarify your methodology for creating these standards to ensure reproducibility and transparency.

Response: We have revised the Methods section to provide a detailed explanation of how the gold standard semantic segmentation masks were manually created for the camera signal, ensuring clarity and reproducibility of our methodology.

Comment 6: Technical Contribution: The manuscript currently employs a UNet model without notable innovations or modifications. To enhance technical value, clearly state and elaborate on novel aspects or significant modifications of your approach compared to existing methodologies.

Response: While we used a standard UNet model for segmenting the camera signal from raw endoscopic data, the primary contribution of our work lies in the application of an AI-based solution to standardize raw endoscopic images by extracting the camera signal. This approach addresses a critical gap in endoscopic research and is novel in its application. Additionally, we have made the trained model weights and methodology publicly available to support the broader research community. Our focus was on applying existing models in innovative ways to solve prevalent challenges in the field, rather than developing a new architecture.

We believe these revisions address all your concerns and have substantially improved the quality of the manuscript. A detailed response to the reviewer comments is also included below.

Once again, we are grateful for the valuable feedback from you and the reviewers, which has significantly enhanced the manuscript.

Dr. Ioannis Kafetzis

Response to Reviewer #1

Comment: The quality of the production is still quite low, say, the last line of the abstract is in improper format, there is no where to find Figure 1 and Figure 2, reference 21 has no publication information.

Response: We appreciate the reviewer’s careful review and apologize for the identified issues. Regarding the last line of the abstract, we resolved the formatting issue related to text justification. We have also ensured that Figures 1 and 2 are now properly referenced and visible in the manuscript. Finally, we have updated reference 21 with the necessary publication information.

Comment: You cannot simply say "Hausdorff distance (HD) is the maximum distance between predictions and the gold standard."

Response: The reviewer is right to point out that the original description of the Hausdorff distance was overly simplistic. To address this, we have now provided a more precise and formal mathematical definition of the Hausdorff distance. Additionally, to make the concept clearer for a broader audience, we included an interpretation of the definition. The revised passage reads:

The method was evaluated by means of intersection over union (IoU) and Hausdorff distance (HD). IoU is defined as the percentage overlapping with the gold standard. The HD, in the context of semantic segmentation, is defined as follows. Let m_1 and m_2 be two binary masks. The one-sided HD from m_1 to m_2 is defined as hd_s (m_1,m_2 )=(max)┬(x∈m_1 )⁡(min)┬(y∈m_2 )⁡〖‖x-y‖_2 〗 . This can be interpreted as the maximum distance from a point x∈m_1 to the closest point y∈m_2. The HD of m_1 and m_2 is defined as the maximum of the two one-sided HDs, that is HD(m_1,m_2 )=(max)┬⁡〖(hd_s (m_1,m_2 ),hd_s (m_2,m_1 )〗. Thus, the HD between the two masks can be seen as the greatest distance from a point in either mask to its nearest point in the other mask.

Comment: The design of the method in the manuscript is not clear how you generate the labels/gold standard for training.

Response: We appreciate the reviewer’s observation that the method for generating the gold standard was not sufficiently detailed. To clarify, we have expanded the relevant section in the Methods to explain that the gold standard semantic segmentation masks were manually created for the camera signal. The revised text reads:

Gold standard binary semantic segmentation masks for each image were manually created to delineate the camera signal– where a value of 1 indicated pixel inclusion within the camera signal and 0 otherwise.

Comment: More importantly, the manuscript simply applied the UNet rather than developed some new techniques, which show very little technical contributions.

Response: While it is true that the paper utilizes a UNet architecture to segment the camera signal from raw endoscopic data, the primary contribution of the manuscript lies in applying AI-based methods to standardize raw endoscopic images. Our approach provides an applicable, practical solution for the research community, and the trained model weights, along with the method itself, are made publicly available to facilitate further research. The intention was not to propose a new architecture but to leverage existing, well-established methods to address a key challenge in the field. We believe this makes a meaningful contribution to the standardization of endoscopic data. We have updated the relevant parts of the abstract, introduction and discussion to highlight our aim and most importantly, the fact that the proposed method and model weights are publicly available.

Response to Reviewer #2

We are thankful to the reviewer for the valuable insights they have provided. We are glad that they support publication of the work.

---

## [Decision Letter · Decision Letter 2]

Advancing artificial intelligence applicability in endoscopy through source-agnostic camera signal extraction from endoscopic images

PONE-D-25-03664R2

Dear Dr. Kafetzis,

We’re pleased to inform you that your manuscript has been judged scientifically suitable for publication and will be formally accepted for publication once it meets all outstanding technical requirements.

Kind regards,

Kazunori Nagasaka

Academic Editor

PLOS ONE

Additional Editor Comments (optional):

Dear Authors,

I am pleased to inform you that your manuscript entitled "Advancing artificial intelligence applicability in endoscopy through source-agnostic camera signal extraction from endoscopic images" (Manuscript ID PONE-D-25-03664R2) has been reviewed carefully and is now accepted for publication in PLOS ONE.

The editorial team and reviewers acknowledge your diligent effort in addressing all the comments and concerns raised during the review process. Your revisions have significantly enhanced the manuscript’s clarity, depth, and scientific rigor, aligning it fully with the standards of our journal.

Again, Thank you for choosing Plos One for your publication.

Sincerely,

Kazunori Nagasaka, M.D., Ph.D.

Academic Editor

PLOS ONE

Reviewers' comments:

Reviewer's Responses to Questions

**Comments to the Author**

Reviewer #2: All comments have been addressed

2. Is the manuscript technically sound, and do the data support the conclusions?

Reviewer #2: Yes

3. Has the statistical analysis been performed appropriately and rigorously?

Reviewer #2: Yes

4. Have the authors made all data underlying the findings in their manuscript fully available?

Reviewer #2: Yes

5. Is the manuscript presented in an intelligible fashion and written in standard English?

Reviewer #2: Yes

Reviewer #2: The authors all reply my concerns. The manuscript technically sound, and do the data support the conclusions. This article can be accepted by PLOS One.

**Do you want your identity to be public for this peer review?** For information about this choice, including consent withdrawal, please see our Privacy Policy

Reviewer #2: **Yes: ** Hsiang-Chen Wang

---

## [Editor Report · Acceptance letter]

PONE-D-25-03664R2

PLOS ONE

Dear Dr. Kafetzis,

I'm pleased to inform you that your manuscript has been deemed suitable for publication in PLOS ONE. Congratulations! Your manuscript is now being handed over to our production team.

Kind regards,

on behalf of

Professor Kazunori Nagasaka

Academic Editor

PLOS ONE